# Predictors of DAPSA Response in Psoriatic Arthritis Patients Treated with Apremilast in a Retrospective Observational Multi-Centric Study

**DOI:** 10.3390/biomedicines11020433

**Published:** 2023-02-02

**Authors:** Andrea Becciolini, Simone Parisi, Patrizia Del Medico, Antonella Farina, Elisa Visalli, Aldo Biagio Molica Colella, Federica Lumetti, Rosalba Caccavale, Palma Scolieri, Romina Andracco, Francesco Girelli, Elena Bravi, Matteo Colina, Alessandro Volpe, Aurora Ianniello, Maria Chiara Ditto, Valeria Nucera, Veronica Franchina, Ilaria Platè, Eleonora Di Donato, Giorgio Amato, Carlo Salvarani, Simone Bernardi, Gianluca Lucchini, Francesco De Lucia, Francesco Molica Colella, Daniele Santilli, Natalia Mansueto, Giulio Ferrero, Antonio Marchetta, Eugenio Arrigoni, Rosario Foti, Gilda Sandri, Vincenzo Bruzzese, Marino Paroli, Enrico Fusaro, Alarico Ariani

**Affiliations:** 1Internal Medicine and Rheumatology Unit, Department of Medicine, University Hospital of Parma, 43121 Parma, Italy; 2Rheumatology Unit, Department of General and Specialistic Medicine, Azienda Ospedaliera Universitaria, Città della Salute e della Scienza di Torino, 10121 Turin, Italy; 3Rheumatology Outpatient Clinic, Internal Medicine Unit, Civitanova Marche Hospital, 62012 Civitanova Marche, Italy; 4Internal Medicine Unit, Rheumatology Outpatient Clinic, Ospedale “A. Murri”, 63900 Fermo, Italy; 5Rheumatology Unit, Policlinico San Marco University Hospital of Catania, 95121 Catania, Italy; 6Rheumatology Unit, Azienda Ospedaliera Papardo, 98158 Messina, Italy; 7Rheumatology Unit, Azienda USL of Modena and AOU Policlinico of Modena, 41100 Modena, Italy; 8Department of Medical-Surgical Sciences and Biotechnologies, Sapienza University of Rome, Polo Pontino, 04100 Latina, Italy; 9Unit of Internal Medicine and Rheumatology, “Nuovo Regina Margherita/S. Spirito” Hospital, ASL Roma 1, 00153 Rome, Italy; 10Distretto Socio Sanitario ASL 1 Imperiese, 18100 Imperia, Italy; 11Rheumatology Service, Internal Medicine Unit, GB Morgagni Hospital, 47121 Forli, Italy; 12Rheumatology Unit, Ospedale Guglielmo da Saliceto, 29121 Piacenza, Italy; 13Rheumatology Service, Section of Internal Medicine, Department of Medicine and Oncology, Ospedale Santa Maria della Scaletta, 40026 Imola, Italy; 14Alma Mater Studiorum, Department of Biomedical and Neuromotor Sciences, University of Bologna, 40126 Bologna, Italy; 15Rheumatology Unit, IRCCS Sacro Cuore Don Calabria, 37024 Bologna, Italy; 16Rheumatology Outpatient Unit, ASL Novara, 28100 Novara, Italy; 17UOC Oncologia Medica Azienda Ospedaliera Papardo, 98158 Messina, Italy; 18Rheumatology Unit, University of Modena and Reggio Emilia, 41100 Modena, Italy; 19Internal Medicine Unit, University Bicocca Milan, 20126 Milano, Italy; 20Unit of Diagnostic and Interventional Radiology, Santa Corona Hospital, 17027 Pietra Ligure, Italy

**Keywords:** psoriatic arthritis, apremilast, DAPSA

## Abstract

Background: To date, only a few real-world-setting studies evaluated apremilast effectiveness in psoriatic arthritis (PsA). The aims of this retrospective observational study are to report long-term Disease Activity Index for Psoriatic Arthritis (DAPSA) response of apremilast in PsA patients and to analyze the predictors of clinical response. Methods: All PsA consecutive patients treated with apremilast in fifteen Italian rheumatological referral centers were enrolled. Anamnestic data, treatment history, and PsA disease activity (DAPSA) at baseline, 6 months, and 12 months were recorded. The Mann–Whitney test and chi-squared tests assessed the differences between independent groups, whereas the Wilcoxon matched pairs signed-rank test assessed the differences between dependent samples. Logistic regressions verified if there were factors associated with achievement of DAPSA low disease activity or remission at 6 and 12 months. Results: DAPSA low disease activity or remission rates at 6 and 12 months were observed, respectively, in 42.7% (*n* = 125) and 54.9% (*n* = 161) patients. Baseline DAPSA was inversely associated with the odds of achieving low disease activity or remission at 6 months (odds ratio (OR) 0.841, 95% confidence interval (CI) 0.804–0.879; *p* < 0.01) and at 12 months (OR 0.911, 95% CI 0.883–0.939; *p* < 0.01). Conclusions: Almost half of the PsA patients receiving apremilast achieved DAPSA low disease activity or remission at 6 and 12 months. The only factor associated with achievement of low disease activity or remission at both 6 and 12 months was baseline DAPSA.

## 1. Introduction

Psoriatic arthritis (PsA) is a chronic inflammatory disease that affects joints, tendons, and entheses, that can lead to progressive and destructive joint damage and functional disability [1]. Apremilast is an oral inhibitor of phosphodiesterase 4 that EULAR guidelines recommend to use in PsA patients with moderate activity or when other agents are contraindicated [2]. In four phase III studies, PALACE 1, PALACE 2, PALACE 3, and PALACE 4, apremilast demonstrated a significant clinical response in patients with PsA that were naïve or experienced with other disease-modifying antirheumatic drugs (DMARDs) [3,4,5,6]. A recent post hoc analysis of the PALACE 1–3 studies highlighted that 131/375 (39.4%) PsA patients achieved a Clinical Disease Activity Index for Psoriatic Arthritis (cDAPSA) low disease activity or remission at 52 weeks of treatment with apremilast [7]. Moreover, the probability of achieving low disease activity or remission was greater for patients with baseline moderate versus high disease activity [7]. Similarly, a post hoc analysis of the PALACE 4 study reported that 62/138 (44.9%) PsA patients achieved a cDAPSA low disease activity or remission at 52 weeks of treatment with apremiliast, with higher probability for those with baseline moderate disease activity [8].

Despite this evidence from registration and post hoc analysis studies, the external validity of randomized controlled trials is significantly hindered by stringent inclusion and exclusion criteria, thus, limiting the generalizability to real-world clinical practice [9]. Only a few papers have addressed apremilast effectiveness in a real-word setting; however, they reported prevalently short-term results in relatively small populations [10,11,12,13,14]. Therefore, to date, real-word data reporting apremilast effectiveness are lacking.

Recently, we analyzed the 3-year apremilast retention rate, reasons for discontinuation, and factors related to treatment persistence in a large multicenter observational retrospective study [15].

The aims of the present study, including the same large multicenter observational retrospective cohort, are to report the baseline characteristics of PsA patients according to their possible inclusion in the PALACE trials, the long-term DAPSA response of apremilast, and the predictors of clinical response.

## 2. Materials and Methods

### 2.1. Patients

The analyzed population is part of the BIRRA (Biologics Retention Rate Assessment) project, an observational retrospective study [15]. All PsA consecutive patients from fifteen Italian rheumatological referral centers were screened. Inclusion criteria were as follows: (a) PsA diagnosis according to CASPAR criteria [16], (b) apremilast prior to actual use, (c) availability of data about treatment beginning and discontinuation, and (d) availability of data regarding DAPSA at baseline, 6 months, and 12 months. Patients who received apremilast and bDMARDs at the same time or only for dermatologic indication (i.e., psoriasis (PsO)) were excluded.

The inclusion/exclusion criteria of the PALACE 1–2 studies [3,4] divided the cohort into two groups. Subjects who satisfied these criteria made up the PALACE-like subgroup (PLG); the other group was the real-world subgroup (RWG).

### 2.2. Data

For each patient, the following data were recorded: general characteristics (age, sex, body mass index (BMI), smoking habit, presence of the human leukocyte antigen (HLA) class I molecule B27, PsA and PsO onset, and diagnosis date), PsA phenotype, as judged by the treating physician (oligo-articular (i.e., less than 5 swollen joints involved), poli-articular, enthesitic, axial, and dactylic subtype), apremilast-related information (date of the first and last intake), other PsA treatment history (both csDMARDs and bDMARDs), PsA disease activity (number of tender/swollen joints, painful enthesis and fingers affected by dactylitis, C-reactive protein, pain Visual Analog Scale, and patient global assessment values) at baseline, 6 months, and 12 months, and the presence of comorbidities. DAPSA assessed the PsA disease activity [17].

Cancer, HBV, HCV, latent tuberculosis (TB), and other chronic infections were considered as relevant comorbidities. The classification of apremilast treatment interruptions included primary or secondary failure, gastro-intestinal intolerance, neurologic side effects, infection, and cancer.

### 2.3. Statistical Analysis

The D’Agostino–Pearson test verified the variables’ normal distribution. Continuous variables were reported as median value and interquartile range (IQR) and categorical values as percentage.

The Mann–Whitney test and chi-squared tests assessed the differences between PLG and RWG. The Wilcoxon matched pairs signed-rank test assessed the differences between dependent samples. Logistic regressions verified if there were factors (age, sex, BMI, smoking habit, relevant comorbidity, PsA disease duration, baseline DAPSA score, concomitant csDMARDs treatment, number of previous bDMARDs, number of previous cDMARDs) associated with achievement of DAPSA and cDAPSA low disease activity or remission at 6 and 12 months. We performed univariate logistic regression analysis on all variables and included those with a *p*-value < 0.1 into multivariate logistic regression analysis to determine independent prognostic factors of DAPSA low disease activity or remission at 6 and 12 months. The effectiveness outcomes were analyzed with the intention to treat approach, with last observation carried forward where appropriate. A *p*-value < 0.05 was considered statistically significant. Statistical analysis was performed using an online application (www.statskingdom.com, last visit 20 December 2022).

## 3. Results

### 3.1. Patient Characteristics

Among the three-hundred fifty-six PsA patients included in our previous report [15], two-hundred ninety-three with a follow-up period of at least one year were enrolled in the study. One-hundred twenty-six (43%) patients were male. Their median age was 60 (IQR 53–68) years, whereas the median PsA disease duration was 49 (IQR 17–98.25) months. The median DAPSA score at baseline was 24.4 (IQR: 19.4–32). Sixty-nine (23.5%) patients did not receive any DMARDs before apremilast. Relevant comorbidities affected one-hundred thirty-seven (46.8%) patients. The main baseline patient characteristics, according to PLG and RWG, are reported in Table 1.

### 3.2. DAPSA Response

The median DAPSA score in the overall cohort of patients was significantly reduced from baseline at both 6 and 12 months, respectively, 15.8 (IQR 11–23.8, *p* < 0.01) and 14 (IQR 9.3–21.8, *p* < 0.01). Similarly, the median DAPSA score in the PLG and RWG of patients was significantly reduced from baseline at both 6 months, 18.2 (IQR 14.5–23.9, *p* < 0.01) and 15 (IQR 10.1–23.5, *p* < 0.01), and 12 months, 15.5 (IQR 12.1–23.4, *p* < 0.01) and 13.3 (IQR 9.2–21.2, *p* < 0.01). The median DAPSA reduction in PLG and RWG was similar at both 6 and 12 months, respectively −9 (IQR −1.5–−17.5) vs. −8.1 (IQR −2–−12.3), *p* = 0.15, and −13 (IQR −1.5–−20.2) vs. −11 (IQR −2–−15.3), *p* = 0.15 (Figure 1).

In the overall population, the DAPSA remission rates at 6 and 12 months were, respectively, 2.4% (*n* = 7) and 7.2% (*n* = 21). Moreover, in the overall population, the DAPSA low disease activity or remission rates at 6 and 12 months were, respectively, 42.7% (*n* = 125) and 54.9% (*n* = 161). The rates of DAPSA remission, low disease activity, moderate disease activity, and high disease activity at 6 and 12 months are reported in Figure 2. The rates of cDAPSA remission, low disease activity, moderate disease activity, and high disease activity are reported in Appendix A. The DAPSA remission and low disease activity or remission rates were lower in the PLG than in the RWG at 6 months, respectively, 0% (*n* = 0) vs. 2.9% (*n* = 7) (*p* = 0.2) and 22.2% (*n* = 12) vs. 47.3% (*n* = 113) (*p* < 0.01), and at 12 months, respectively, 3.7% (*n* = 2) vs. 7.9% (*n* = 19) (*p* = 0.24) and 42.6% (*n* = 23) vs. 57.7% (*n* = 138) (*p* = 0.04). The rates of DAPSA remission, low disease activity, moderate disease activity, and high disease activity in PLG and RWG at 6 and 12 months are reported in Figure 3.

### 3.3. Predictors of DAPSA Low Disease Activity or Remission at 6 and 12 Months 

Logistic regressions were performed to assess the relationship between age, BMI, sex, smoking habit, relevant comorbidity, PsA duration, baseline disease activity and concomitant csDMARDs, number of previous bDMARDs, number of previous cDMARDs, and the achievement of low disease activity or remission at 6 and 12 months. At both time points, lower baseline DAPSA was associated with the achievement of low disease activity or remission at 6 and 12 months. In particular, baseline DAPSA was inversely associated with the odds of achieving low disease activity or remission at 6 months (Odds Ratio (OR) 0.841, 95% Confidence Interval (CI) 0.804–0.879; *p* < 0.01) and at 12 months (OR 0.911, 95% CI 0.883–0.939; *p* < 0.01). Moreover, a higher number of previous csDMARDs was slightly associated with the odds of achievement of DAPSA low disease activity or remission at 12 months (OR 1.342, 95% CI 1.003–1.795, *p* = 0.048). Table 2 and Table 3 report the univariate and multivariate logistic regressions of variables associated with achievement of DAPSA low disease activity or remission at 6 and 12 months, respectively. Appendix A report the univariate and multivariate logistic regressions of variables associated with achievement of cDAPSA low disease activity or remission at 6 and 12 months, respectively.

## 4. Discussion

To the best of our knowledge, this is the largest study reporting apremilast DAPSA response in a real-world setting PsA cohort. Although our patients do not derive from a national registry, the fifteen centers that participated in this study are well distributed on the Italian territory; therefore, they may represent the country-prescribing scenario. 

The first result of our study highlighted that less than one-fifth of the PsA patients treated with apremilast in a real-world setting satisfied the inclusion/exclusion criteria of the PALACE studies. When comparing the RWG and the PLG, there are some striking differences in baseline characteristics, such as disease activity, disease duration, and comorbidity burden. Therefore, we can conclude that real-world data have a pivotal role in characterizing the effectiveness of a treatment.

The second result of our study reported the 6- and 12-month DAPSA response rates of PsA patients treated with apremilast. In particular, the DAPSA low disease activity or remission rates at 6 and 12 months were, respectively, 42.7% and 54.9%, with significantly higher rates for the RWG in comparison to the PLG. Only a few previous studies documented DAPSA response in real-world patients treated with apremilast [10,11,12,13]. A previous Italian multicentric study reported a lower low disease activity or remission rate at 6 months (29.2%) when compared to our cohort (42.7%); however, it is noteworthy that authors analyzed a smaller population with a short period of follow-up [10]. A direct comparison with the other real-world studies is not possible since they did not report DAPSA response; however, improvements in symptoms were described in 43.5–60.9% of patients [12,13]. Two recent post hoc analyses of the PALACE 1–3 studies and PALACE 4 study reported 52 weeks of cDAPSA low disease activity or remission, respectively, in 39.4% and 44.9% of PsA patients treated with apremilast [7,8]. When comparing these rates to the one reported in our cohort of patients, it appears to be similar, although slightly higher (54.9%). However, it should be pointed out that if we consider only the PLG, the DAPSA low disease activity or remission at 12 months is comparable (42.6%). Interestingly, both in the PLG and the RWG, we observed a similar DAPSA decrease at 6 and 12 months from baseline. 

Finally, we analyzed the factors associated with achievement of DAPSA low disease activity or remission at 6 and 12 months. We found that at both time points, higher baseline DAPSA was significantly associated with reduced probability of achieving low disease activity or remission. To the best of our knowledge, similar analysis in a real-world setting has been reported only by the RAPPER study [10]. In that study, female sex was a strong negative predictor of DAPSA response; however, the authors analyzed only a shorter-term endpoint (3 months) and baseline disease activity was not evaluated. On the other hand, similarly to what was observed in our cohort of PsA patients, the post hoc analyses of PALACE 1–3 studies and PALACE 4 study clearly highlighted that the probability of achieving cDAPSA low disease activity or remission at week 52 was greater for patients with moderate versus high disease activity at baseline [7,8].

The main strength of our study is that it assessed a large group of PsA patients treated with apremilast in a real-world setting. On the other hand, our study has some limitations. The main limitation of our study lies in its observational retrospective design, leading to a possible selection bias stemming from including patients with different treatment response odds. Another limitation of this study is the relatively short follow-up period, since we did not analyze further time points in addition to 12 months. Furthermore, given the retrospective nature of the study, we were not able to assess different composite disease activity indexes other than DAPSA, and to evaluate other variables, such as corticosteroid or NSAID use. Finally, it is worth mentioning that in Italy, in addition to the national guidelines of treatment, rheumatologists must also comply with regional provisions, which can be very different from center to center and that could have partially driven the choice of apremilast use. 

## 5. Conclusions

In conclusion, our real-world retrospective multicentric study highlighted that the majority of PsA patients treated with apremilast differs from the ones treated in registration randomized controlled trials. This result confirms the relevance of observational studies in assessing the effectiveness of treatments. Secondly, we reported that DAPSA low disease activity or remission rates of apremilast in a real-world setting of PsA patients were comparable to the ones reported in registration trials. Finally, we confirmed that even the major factor associated with achievement of DAPSA low disease activity or remission was baseline disease activity. Further studies, especially in patients treated for a longer duration with apremilast, should be encouraged in order to confirm our results.

## Figures and Tables

**Figure 1 biomedicines-11-00433-f001:**
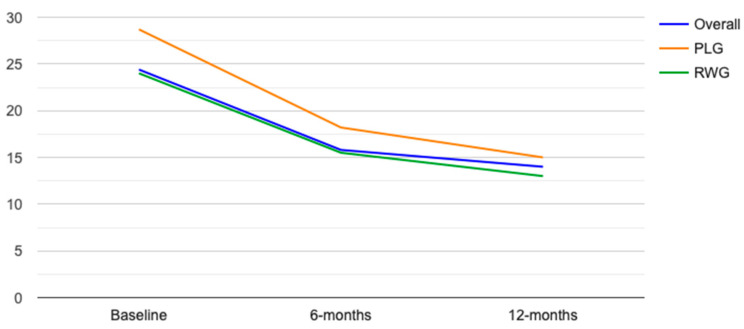
DAPSA median values at baseline, 6 months, and 12 months in overall cohort (blue line), PLG (orange line), and RWG (green line).

**Figure 2 biomedicines-11-00433-f002:**
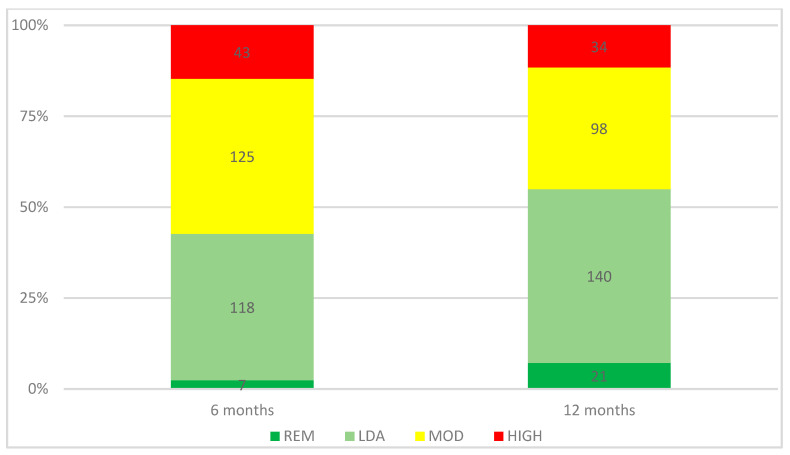
DAPSA scores at 6 and 12 months. REM remission, LDA low disease activity, MOD moderate disease activity, HIGH high disease activity.

**Figure 3 biomedicines-11-00433-f003:**
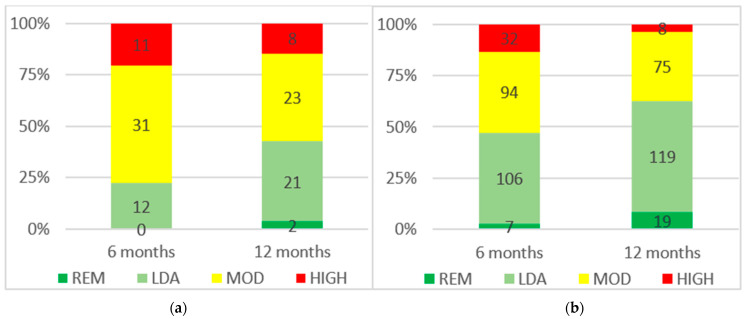
DAPSA scores at 6 and 12 months in PLG (**a**) and RWG (**b**). REM remission, LDA low disease activity, MOD moderate disease activity, HIGH high disease activity.

**Table 1 biomedicines-11-00433-t001:** Baseline characteristics of PsA patients.

		Total Cohort	PLG	RWG
*n*		293	54	239
M:F		126:167	21:33	105:134
Age, median (IQR), yrs		60 (53–68)	61 (52.5–67)	60 (53–68)
Smokers: yes/former/no *		46:37:208	10:9:35	36:28:173
BMI, median (IQR), kg/m^2^ **		26 (23.7–29.4)	25.9 (23.5–29.7)	26.1 (23.8–29.3)
PsA duration, median (IQR), months		49 (17–98.25)	23 (13–65)	53 (20–101.5)
PsA phenotype	Oligo-articular Poli-articular Enthesitic Dactylitis Axial	164 129 144 101 36	26 28 23 18 7	138 101 121 83 29
SJC, median (IQR)		3 (2–4)	4 (3–6)	2 (2–4)
TJC, median (IQR)		6 (3–10)	6 (4–12)	6 (3–9)
LEI, median (IQR)		0 (0–2)	0 (0–2)	0 (0–2)
Dactylitis, median (IQR), fingers		0 (0–1)	0 (0–0)	0 (0–1)
CRP, median (IQR), mg/dL		2.0 (0.7–4.6)	2.5 (1.0–5.2)	1.9 (0.6–4.2)
DAPSA, median (IQR)		24.4 (19.4–32)	28.7 (23.1–36.9)	24 (18.5–31)
Prior csDMARDs use, *n*	MTX LFN SSZ CYA	188 50 76 27	32 8 14 1	156 42 62 26
Prior bDMARDs use, *n*	TNFi IL17i IL12/IL23i Abatacept	81 21 20 2	0 0 0 0	81 21 20 2
Concomitant csDMARDs, *n*		58	10	48
Concomitant relevant disease, *n*	Cancer HCV/HBV Latent TB Other infections	89 16 17 20	0 0 0 3	89 16 17 17

Data missing in 2 (*) and 20 (**) patients. BMI body mass index, PLG PALACE-like group, RWG real-world group, IQR interquartile range, SJC swollen joint count, TJC tender joint count, LEI Leeds enthesitis index, CRP C-reactive protein, DAPSA Disease Activity index for Psoriatic Arthritis.

**Table 2 biomedicines-11-00433-t002:** Univariate and multivariate analysis of variables associated with achievement of DAPSA low disease activity and remission at 6 months.

Univariate Analysis				Multivariate Analysis		
Variable	OR	95% CI	*p*	OR	95% CI	*p*
Age	0.988	0.969–1.007	0.943			
Smoke	1.226	0.646–2.325	0.533			
Sex	1.136	0.712–1.813	0.592			
BMI	0.985	0.934–1.039	0.583			
Relevant comorbidity	1.22	0.767–1.941	0.4			
Disease duration	1	0.997–1.004	0.87			
Baseline DAPSA	0.837	0.8–0.876	<0.0001	0.841	0.804–0.879	<0.0001
Concomitant csDMARD	0.652	0.358–1.187	0.161			
Number of previous bDMARD	0.718	0.578–0.893	0.0029	0.777	0.604–1.0003	0.0502
Number of previous csDMARD	1.173	0.909–1.512	0.219			

OR odds ratio, CI confidence interval, BMI body mass index, DAPSA Disease Activity index for PSoriatic Arthritis.

**Table 3 biomedicines-11-00433-t003:** Univariate and multivariate analysis of variables associated with achievement of DAPSA low disease activity and remission at 12 months.

Univariate Analysis				Multivariate Analysis		
Variable	OR	95% CI	*p*	OR	95% CI	*p*
Age	0.999	0.98–1.018	0.906			
Smoke	1.181	0.621–2.247	0.613			
Sex	1.309	0.820–2.087	0.259			
BMI	0.980	0.929–1.033	0.457			
Relevant comorbidity	1.299	0.818–2.064	0.267			
Disease duration	1.002	0.998–1.006	0.289			
Baseline DAPSA	0.908	0.881–0.937	<0.0001	0.911	0.883–0.939	<0.0001
Concomitant csDMARD	1.577	0.871–2.856	0.133			
Number of previous bDMARD	0.84	0.691–1.022	0.0815	0.856	0.691–1.061	0.156
Number of previous csDMARD	1.328	1.021–1.726	0.034	1.342	1.003–1.795	0.048

OR odds ratio, CI confidence interval, BMI body mass index, DAPSA Disease Activity index for Psoriatic Arthritis.

## Data Availability

Data can be made available from the corresponding author upon request.

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
