# Peer review of "Predictors of DAPSA Response in Psoriatic Arthritis Patients Treated with Apremilast in a Retrospective Observational Multi-Centric Study"

_biomedicines, 2023, doi:10.3390/biomedicines11020433_

Round 1
Reviewer 1 Report
The study submitted for review concerned the predictors of DAPSA response in psoriatic arthritis patients treated with apremilast. The study was well-designed and described. A vital point is the large number of centres participating in the study. Apremilast is not the drug of first choice in the case of high disease activity in PSA, however, it is applicable in the specific cases described by the authors. Few papers on the effectiveness of apremilast treatment in everyday clinical practice have been published so far, so the presented study provides valuable information for clinicians.
Minor comment:
1. Please indicate whether the DAPSA response depended on the disease's clinical form.
I have no other significant comments.
Author Response
Thank you for your revision
- Please indicate whether the DAPSA response depended on the disease's clinical form.
- At both time points (6 and 12 months) the oligoarticular subtype is associated with and increased probability of achieving DAPSA low disease activity or remission. However, oligoarticular patients showed, as expected, lower DAPSA values. Therefore, the inclusion of oligoarticular subtype in the analysis can be misleading. There were no significative association between enthesitic, dactilytis and axial subtype and DAPSA low disease activity or remission at both 6 and 12 months. Given the lack of possible association between these variables, we did not include in the final analysis.
Reviewer 2 Report
This is a multicenter retrospective observational study in patients starting apremilast for the systemic management of psoriatic arthritis (PsA). The objectives of the work were (i) to analyze 6 and 12 months DAPSA response rates and (ii) the predictors of clinical response. As the main result of their work, the authors reported that half patients achieved DAPSA low disease activity or remission over-time, and baseline DAPSA associated with achievement of low disease activity or remission. Given the relative paucity of real-life studies on this topic, this work is valuable since it depicts the real-life usefulness of this drug across several centers in Italy. However, several points should be clarified.
Major comments:
The authors focused on a peripheral joints-based outcome measure. This is important since the majority of patients in their cohort suffered from peripheral joint disease (Table 1); however, it is well-known that DAPSA has several limitations, and one of the most important relates to the absence of consideration for other significant domains for a disease like PsA, e.g. enthesitis. Were the authors able to calculate other disease activity indexes, like PASDAS as an example, in line with other real-life studies or registries (i.e. https://doi.org/10.1186/s13063-021-05142-7)?
In the introduction, the authors named two post-hoc analyses of the PALACE program focusing on cDAPSA remission. However, the authors focused on DAPSA instead of cDAPSA. I suggest performing a secondary analysis to evaluate if the predictors of cDAPSA were different from the predictors of DAPSA remission or low-disease activity.
Since it is well-known that, despite not recommended, many patients with PsA receive glucocorticoids (GCs) in real-life setting, did the authors collect information regarding previous and ongoing GCs administration and dosage? Similarly, they should report information regarding previous and ongoing NSAIDs assumption.
“Age, sex, BMI, smoke habit, relevant comorbidity, PsA disease duration, baseline DAPSA score and concomitant csDMARDs treatment” were considered as possible predictors of DAPSA response at 12 months in logistic regression analysis. How were these variable chosen? Did the authors apply univariate analysis and then multivariate analyses (e.g. using backward stepwise method)? Please, specify. If the variables were pre-specified, I suggest considering also the number of previous cs- and b-DMARDs as a possible predictor.
Predictors of DAPSA low disease activity or remission at 6 and 12 months: the authors should include a table or a figure reporting ORs and 95%CI for all the variables tested in the model, since I did not find it in the manuscript.
“Materials and methods”: In the Results section, the authors reported “Two-hundred ninety-three PsA patients with a follow-up period of at least one year were enrolled in the study”. It is unclear from the “Materials and Methods” section if patients enrolled in the study should have had at least 12 months of follow up available while taking apremilast, or if the authors enrolled patients starting apremilast with 12 months of follow up available, independently from the continuation of the study drug during the study period. This point needs to be well specified, since: (i) considering only persistent apremilast-users, the authors have selected a population of persistent-in-treatment patients, that had benefits from the drug or at least tolerate it, and this impacts on the primary objective of the study (e.g. rates of LDA/remission achievement); (ii) it is expected that some patients have interrupted the study drug earlier than the pre-specified time-points, and if this second is the case, the authors should describe the reasons for earlier interruption, and perform survival curves with cox proportional hazard models to evaluate the predictors of drug discontinuation. This is a relevant point in my opinion.
The authors reported many baseline and treatment-related variables subdivided in two main groups: PLG (a) and RWG (b). However, this point was not stated among the objectives of the study. Please, homogenize.
Minor comments:
Title: The design of the study (retrospective observational) should be underlined also in the title.
Abstract: “A p-value<0.05 was considered statistically significant”. I would remove this sentence from the abstract.
Introduction: “Apremilast is an oral inhibitor of the phosphodiesterase 4 that EULAR guidelines recommend to use in PsA patients with moderate active or when other agents are contraindicated”. There is a typo.
Introduction: “In three phase III studies PALACE 1, PALACE 2. PALACE 3 and PALACE 4…”. Please, homogenize. The real number should be 3 or 4.
Materials and Methods: “PALACE-like subgroup (PLG)”. Please, underline which criteria were considered to be eligible in this subgroup.
Materials and methods: “PsA phenotype (oligo-articular, poli-articular, enthesitic, axial, and dactylics subtype)”. Which definition was adopted for phenotype assessment? Please, specify.
I suggest revising the manuscript with a native English, since many typos and ortographic errors are present in the text.
Author Response
Thank you for your comments. We think that your suggestions considerably improved the manuscript quality.
Major comments:
The authors focused on a peripheral joints-based outcome measure. This is important since the majority of patients in their cohort suffered from peripheral joint disease (Table 1); however, it is well-known that DAPSA has several limitations, and one of the most important relates to the absence of consideration for other significant domains for a disease like PsA, e.g. enthesitis. Were the authors able to calculate other disease activity indexes, like PASDAS as an example, in line with other real-life studies or registries (i.e. https://doi.org/10.1186/s13063-021-05142-7)?
- Unfortunately, due to retrospective design of the study we are not able to calculate other disease activity indexes.
In the introduction, the authors named two post-hoc analyses of the PALACE program focusing on cDAPSA remission. However, the authors focused on DAPSA instead of cDAPSA. I suggest performing a secondary analysis to evaluate if the predictors of cDAPSA were different from the predictors of DAPSA remission or low-disease activity.
- As suggested by the reviewer we performed an analysis to evaluate the cDAPSA low disease activity or remission predictors at 6 and 12 months. At both time points the rates of low disease activity or remission were similar (6 months, n=142; 12 months, n=173). The univariate and multivariate analyses for cDAPSA low disease activity or remission predictors were reported in the following tables (1 and 2). The results of both analyses confirms that baseline DAPSA is the strongest predictor of achievement of cDAPSA low disease activity or remission predictors. Interestingly, at six months two other variables appear to be significantly associated with achievement of cDAPSA low disease activity or remission: presence of relevant comorbidity and number of previous bDMARDs. We can agree that these findings are interesting and sound, however given that these results are significant only at the six months time point for the cDAPSA analysis we suspect that these findings could only be driven by chance.
Table 1. Univariate and multivariate analysis of variables associated with achivement of cDAPSA low disease activity and remission at 6 months.
|
Univariate Analysis |
Multivariate Analysis |
|||||
|
Variable |
OR |
95% CI |
p |
OR |
95% CI |
p |
|
Age |
0.987 |
0.968 – 1.006 |
0.799 |
|
|
|
|
Smoke |
1.039 |
0.549 – 1.968 |
0.905 |
|
|
|
|
Sex |
1.178 |
0.741 – 1.871 |
0.488 |
|
|
|
|
BMI |
0.991 |
0.941 – 1.044 |
0.738 |
|
|
|
|
Relevant comorbidity |
1.52 |
0.958 – 2.412 |
0.075 |
1.943 |
1.132 – 3.336 |
0.016 |
|
Disease duration |
0.999 |
0-996 – 1.003 |
0.688 |
|
|
|
|
Baseline DAPSA |
0.883 |
0.852 – 0.915 |
<0.0001 |
0.881 |
0.849 – 0.913 |
<0.0001 |
|
Concomitant csDMARD |
0.764 |
0.428 – 1.364 |
0.362 |
|
|
|
|
Number of previous bDMARD |
0.711 |
0.564 – 0.895 |
0.0036 |
0.779 |
0.614 – 0.989 |
0.0407 |
|
Number of previous csDMARD |
1.046 |
0.814 – 1.344 |
0.726 |
|
|
|
Table2. Univariate and multivariate analysis of variables associated with achivement of cDAPSA low disease activity and remission at 12 months.
|
Univariate Analysis |
Multivariate Analysis |
|||||
|
Variable |
OR |
95% CI |
p |
OR |
95% CI |
p |
|
Age |
1.007 |
0.9877 – 1.027 |
0.887 |
|
|
|
|
Smoke (yes) |
0.972 |
0.510 – 1.852 |
0.932 |
|
|
|
|
Sex |
1.305 |
0.813 – 2.095 |
0.269 |
|
|
|
|
BMI |
0.986 |
0.935 – 1.039 |
0.598 |
|
|
|
|
Relevant comorbidity |
1.337 |
0.837 – 2.137 |
0.224 |
|
|
|
|
Disease duration |
1.002 |
0.998 – 1.005 |
0.359 |
|
|
|
|
Baseline DAPSA |
0.933 |
0.908 – 0.959 |
<0.0001 |
0.935 |
0.909 – 0.961 |
<0.0001 |
|
Concomitant csDMARD |
1.283 |
0.708 – 2.325 |
0.412 |
|
|
|
|
Number of previous bDMARD |
0.808 |
0.664 – 0.984 |
0.0336 |
0.843 |
0.688 – 1.034 |
0.101 |
|
Number of previous csDMARD |
1.202 |
0.926 – 1.562 |
0.167 |
|
|
|
Since it is well-known that, despite not recommended, many patients with PsA receive glucocorticoids (GCs) in real-life setting, did the authors collect information regarding previous and ongoing GCs administration and dosage? Similarly, they should report information regarding previous and ongoing NSAIDs assumption.
- Unfortunaly, due to the retrospective design of the study we have no data regarding concomitant corticosteroid or NSAIDs use.
“Age, sex, BMI, smoke habit, relevant comorbidity, PsA disease duration, baseline DAPSA score and concomitant csDMARDs treatment” were considered as possible predictors of DAPSA response at 12 months in logistic regression analysis. How were these variable chosen? Did the authors apply univariate analysis and then multivariate analyses (e.g. using backward stepwise method)? Please, specify. If the variables were pre-specified, I suggest considering also the number of previous cs- and b-DMARDs as a possible predictor.
- The analyses of predictors have been modified as requested. In the methods section has been cleared how the univariate and multivariate analyses have been carried out.
Predictors of DAPSA low disease activity or remission at 6 and 12 months: the authors should include a table or a figure reporting ORs and 95%CI for all the variables tested in the model, since I did not find it in the manuscript.
- The tables with OR, 95% CI and p have been added to the manuscript.
“Materials and methods”: In the Results section, the authors reported “Two-hundred ninety-three PsA patients with a follow-up period of at least one year were enrolled in the study”. It is unclear from the “Materials and Methods” section if patients enrolled in the study should have had at least 12 months of follow up available while taking apremilast, or if the authors enrolled patients starting apremilast with 12 months of follow up available, independently from the continuation of the study drug during the study period. This point needs to be well specified, since: (i) considering only persistent apremilast-users, the authors have selected a population of persistent-in-treatment patients, that had benefits from the drug or at least tolerate it, and this impacts on the primary objective of the study (e.g. rates of LDA/remission achievement); (ii) it is expected that some patients have interrupted the study drug earlier than the pre-specified time-points, and if this second is the case, the authors should describe the reasons for earlier interruption, and perform survival curves with cox proportional hazard models to evaluate the predictors of drug discontinuation. This is a relevant point in my opinion.
- We have previously published a paper regarding apremilast retention rate, reason of discontinuation and its predictors (reference 15). The present paper included only patients with a follow-up of at least 12 months (ie only patients that have started treatment at least one year before the database lock) and with DAPSA values repoted. OF the 356 initially evaluated patients we included in the present analyses only 293 that met such criteria. The analysis was performed according to an intention to treat approach with a last obersvation carried forward for those interrupting treatment with apremilast. These points have been clarified in the introduction and methods section.
The authors reported many baseline and treatment-related variables subdivided in two main groups: PLG (a) and RWG (b). However, this point was not stated among the objectives of the study. Please, homogenize.
- The point has been stated among the study objectives.
Minor comments:
Title: The design of the study (retrospective observational) should be underlined also in the title.
- The title has been modified as suggested
Abstract: “A p-value<0.05 was considered statistically significant”. I would remove this sentence from the abstract.
- The sentence has been removed from the abstract
Introduction: “Apremilast is an oral inhibitor of the phosphodiesterase 4 that EULAR guidelines recommend to use in PsA patients with moderate active or when other agents are contraindicated”. There is a typo.
- Thanks for the suggestion.
Introduction: “In three phase III studies PALACE 1, PALACE 2. PALACE 3 and PALACE 4…”. Please, homogenize. The real number should be 3 or 4.
- The sentence has been corrected.
Materials and Methods: “PALACE-like subgroup (PLG)”. Please, underline which criteria were considered to be eligible in this subgroup.
- The inclusion criteria of PALACE 1-2 have been utilized to define the PLG. It has been clarified in the methods section.
Materials and methods: “PsA phenotype (oligo-articular, poli-articular, enthesitic, axial, and dactylics subtype)”. Which definition was adopted for phenotype assessment? Please, specify.
- In accordance with EULAR recommendation for treatment patients with oligoarticular phenotype was defined as having less than 5 swollen joints involved. Poliarticular phenotype was defined as having more than 4 joints involved. These two groups were the only mutually exclusive. The other subgroups were defined by the presence of entheseal, axial or dactylitic involvement.
I suggest revising the manuscript with a native English, since many typos and ortographic errors are present in the text.
- The typos and syntax errors have been corrected.
Round 2
Reviewer 2 Report
The authors answered my comments and, in my opinion, the quality of the manuscript significantly improved; however, some points should be further addressed.
Due to retrospective design of the study, the authors were not able to calculate other disease activity indexes, and similarly no information are available regarding corticosteroid or NSAIDs use. These should be mentioned as limitations of the study in the “Discussion” chapter.
The authors clearly reported the predictors of cDAPSA response in univariate and multivariate analysis, as requested. I suggest including these tables as supplementary materials (adding a secondary objective to the work).
Abstract: “The only factor associated with achievement of low disease activity or remission at 6 and 12 months was baseline DAPSA”. This is not completely clear, since a (slightly) statistically significant OR of achieving DAPSA response at 12 months was obtained for “previous csDMARDs use”, as commented in the “Results” section. I suggest specifying that baseline DAPSA was the only factor associated with remission achievement at both time points.
Since “PsA phenotype” definition was judged by the treating physician (I assume at the time of diagnosis), this should be underlined in the text.
Results: “Logistic regressions were performed to assess the relationship between age, BMI, sex, smoke habit, relevant comorbidity, PsA duration, baseline disease activity and concomitant csDMARDs, and the achievement of low disease activity or remission at 6 and 12 months”. This sentence is not fully correct, since also previous cs- and b-DMARDs were included in the model, as specified in the “Materials and Methods”.
Author Response
Thank you for your valuable suggestions.
Due to retrospective design of the study, the authors were not able to calculate other disease activity indexes, and similarly no information are available regarding corticosteroid or NSAIDs use. These should be mentioned as limitations of the study in the “Discussion” chapter.
- We added in the Discussion, this sentence: "Furthermore, given the retrospective nature of the study, we were not able to assess different composite disease activity indexes other than DAPSA, and to evaluate other variables such as corticosteroid or NSAID use."
The authors clearly reported the predictors of cDAPSA response in univariate and multivariate analysis, as requested. I suggest including these tables as supplementary materials (adding a secondary objective to the work).
- A supplementary file with the tables above mentioned, was added
Abstract: “The only factor associated with achievement of low disease activity or remission at 6 and 12 months was baseline DAPSA”. This is not completely clear, since a (slightly) statistically significant OR of achieving DAPSA response at 12 months was obtained for “previous csDMARDs use”, as commented in the “Results” section. I suggest specifying that baseline DAPSA was the only factor associated with remission achievement at both time points.
- Thank you for this point. Now, the last sentence of the abstract is: "The only factor associated with achievement of low disease activity or remission at both 6 and 12 months was baseline DAPSA."
Since “PsA phenotype” definition was judged by the treating physician (I assume at the time of diagnosis), this should be underlined in the text
- Done as you suggested.
Results: “Logistic regressions were performed to assess the relationship between age, BMI, sex, smoke habit, relevant comorbidity, PsA duration, baseline disease activity and concomitant csDMARDs, and the achievement of low disease activity or remission at 6 and 12 months”. This sentence is not fully correct, since also previous cs- and b-DMARDs were included in the model, as specified in the “Materials and Methods”.
- The first sentence of paragraph 3.3 changed as you proposed.